

# Technical Note: Conversion of Isoprene Hydroxy Hydroperoxides (ISOPOOH) on Metal Environmental Simulation Chamber Walls

Anne-Kathrin Bernhammer[1,2], Martin Breitenlechner[1,*], Frank N. Keutsch[3], Armin Hansel[1,2] and the CLOUD Team[4]

[1]Institute for Ion and Applied Physics, University of Innsbruck, 6020 Innsbruck, Austria
[2]IONICON Analytik GmbH, 6020 Innsbruck, Austria
[3]John A. Paulson School of Engineering and Applied Sciences and Department of Chemistry and Chemical Biology, Harvard University, Cambridge, MA, USA
[4]CERN, Geneva, Switzerland
[*]now at: John A. Paulson School of Engineering and Applied Sciences and Department of Chemistry and Chemical Biology, Harvard University, Cambridge, MA, USA

*Correspondence to*: Armin Hansel (Armin.Hansel@uibk.ac.at)

## Abstract

Sources and sinks of isoprene oxidation products from low NOx isoprene chemistry have been studied at the CERN CLOUD (Cosmics Leaving Outdoor Droplets) chamber with a custom-built selective reagent ion time of flight mass spectrometer (SRI-ToF-MS), which allows quantitative measurement of isoprene hydroxy hydroperoxides (ISOPOOH).

The measured concentrations of the main oxidation products were compared to chemical box model simulations based on the Leeds Master Chemical Mechanism (MCM) v3.3.The modelled ISOPOOH concentrations are by a factor of 20 higher than the observed and methyl vinyl ketone (MVK) and methacrolein (MACR) concentrations are by a factor of up to 2 lower compared to observations, despite the artifact-free detection method.

Addition of catalytic conversion of 1,2-ISOPOOH and 4,3-ISOPOOH to MVK and MACR on the stainless steel surface of the chamber to the chemical mechanism resolves the discrepancy between model predictions and observation. This suggests that isoprene chemistry in a metal chamber under low $NO_x$ conditions cannot be described by a pure gas phase model alone.

Biases in the measurement of ISOPOOH, MVK and MACR can not only be caused intra-instrumentally but also by the general experimental setup.

The work described here extends the role of heterogeneous reactions affection gas phase composition and properties from instrumental surfaces, described previously, to general experimental setups. The role of such conversion reactions on real environmental surfaces is yet to be explored.





## 1 Introduction

Isoprene is the predominant non-methane biogenic volatile organic compound (BVOC) emitted into the atmosphere (Guenther et al., 2006). Due to its high emissions, its chemistry plays an important role in the oxidative processes within the atmosphere which are driven by catalytic cycles of hydrogen oxides ($HO_x$) and nitrogen oxides ($NO_x$), and which are directly
coupled to the formation of secondary organic aerosol (SOA).

Isoprene shows a high reactivity towards the atmospheric oxidants OH and $NO_3$ and a much lower reactivity towards $O_3$. Its globally dominant sink is the reaction with the OH radical (Archibald et al., 2010) since ozonolysis is a comparatively slow process and reaction with $NO_3$ is only relevant at night and during the early evening, when isoprene emissions are lower than during the day (Stroud, 2002; Warneke, 2004). Reaction with OH not only produces a variety of small carbonyls and
hydroxycarbonyls but also, as main first generation products, hydroxyperoxides that retain the $C_5$ carbon backbone. The oxidation pathway which forms the latter is initiated by addition of OH to the isoprene double bond, followed by the reaction of the resulting allylic radical with molecular oxygen ($O_2$). This leads to the formation of α-hydroxyperoxy radicals ($ISOPO_2$) whose subsequent reactions are highly $NO_x$ dependent. Under pristine ($NO_x < 100$ pptv) and $HO_2$ dominant conditions most $ISOPO_2$ radicals react to isoprene hydroxy hydroperoxides (ISOPOOH) and their subsequent oxidation
products isoprene epoxydiols (IEPOX) (Crounse et al., 2013; Crounse et al., 2011; Paulot et al., 2009; St. Clair et al., 2016). In urban, $NO_x$ dominated, regions $ISOPO_2$ reacts with NO to alkoxy radicals, ISOPO, which decompose to form higher volatility products like MVK and MACR and formaldehyde (HCHO), which are also the main products from reaction of isoprene with $O_3$.

The first generation oxidation products from the $HO_x$ and $NO_x$ dominated pathways differ greatly in terms of chemical and
physical properties and hence play different roles in the atmosphere. The vapour pressure of ISOPOOH is about three orders of magnitude lower than that of MVK and MACR and that of IEPOX is even lower, which increases the importance of their partitioning to surfaces and the condensed phase (Rivera-Rios et al., 2014). ISOPOOH and IEPOX have been shown to play an important role in SOA formation and reactive uptake (Bates et al., 2014; Gaston et al., 2014; Krechmer et al., 2015; Nguyen et al., 2014a).

A correct assessment of low $NO_x$ isoprene chemistry is of great importance but the measurement of ISOPOOH has proven to be difficult by means of readily available commercial VOC instrumentation. Most state of the art instruments like commercial proton transfer reaction (time-of-flight) mass spectrometer (PTR-(ToF)-MS) instruments and also gas chromatograph-mass spectrometer (GC-MS) instruments have biases/artifacts when measuring ISOPOOH and IEPOX. ISOPOOH decomposes isomer specific to smaller carbonyls. 1,2-ISOPOOH forms MVK and formaldehyde and 4,3-
ISOPOOH forms MACR and formaldehyde. Decomposition efficiency can vary strongly depending on instrumental settings such as temperature and contact time and type of surface materials, especially transition metal surfaces (Liu et al., 2013; Nguyen et al., 2014b; Rivera-Rios et al., 2014). The intra-instrumental decompositions may lead to a misrepresentation of ambient concentrations.



Recent laboratory studies of 1,2-ISOPOOH, 4,3-ISOPOOH and β-IEPOX standards with a new custom-built SRI-ToF-MS show significantly reduced catalytic decomposition reactions due to a short residence time and the replacement of the metal drift rings with conductive PEEK which allows the discrimination between MVK/MACR and the ISOPOOH isomers (Mentler et al., 2016).

Here we present measurements with the customised SRI-ToF-MS during low $NO_x$ isoprene oxidation experiments aimed at studying low-$NO_x$ SOA formation during the CLOUD 9 campaign (autumn 2014) and compare these with a chemical box model based on the Leeds Master Chemical Mechanism.

## 2 Experimental

### 2.1 The CLOUD Chamber

The experiments were carried out at the CLOUD chamber at CERN during the CLOUD 9 campaign (autumn 2014). The chamber consists of a 26 m³ electro polished stainless-steel cylinder in which a full range of tropospheric conditions can be reproduced. The chamber is described in detail elsewhere (Duplissy et al., 2016; Kirkby et al., 2011). Inflow of pure air, generated from evaporation of cryogenic nitrogen ($N_2$) and oxygen ($O_2$) at a ratio of 79:21, was set to match the instrumental sample flows of 150 standard litres per minute (slpm). Temperature, chamber pressure and relative humidity (RH) were

continuously monitored, as well as the trace gases $O_3$, $NO_2$, $NO_x$ and $SO_2$. Isoprene (100 ppm in $N_2$, CARBAGAS CH) and ozone were continuously injected into the chamber. Organic trace gases were measured with a custom-built selective reagent ionisation time-of-flight mass spectrometer (SRI-ToF-MS). Typical experimental conditions are listed in Table 1.

### 2.2 SRI-ToF-MS

The custom-built SRI-ToF-MS (Breitenlechner and Hansel, 2016) was optimised by replacing the metal drift rings of the

high flow drift tube (here: 800 mL $min^{-1}$ compared to 10-30 mL $min^{-1}$ in a standard instrument) with conductive PEEK (polyether ether ketone) drift rings. These changes were essential to minimize decomposition reaction of ISOPOOH which are observed in standard PTR-ToF-MS instruments (Rivera-Rios et al., 2014). The operation as an SRI instrument offers the high mass resolution of a PTR-ToF-MS (Graus et al., 2010) combined with the capability to separate functional isomers. To achieve this separation, the SRI-ToF-MS utilises the different chemical ionisation pathways of switchable primary reagent

ions (here: $H_3O^+$ and $NO^+$). This allows the identification of possible interferences, e.g., due to fragmentation onto the same mass to charge ratio in $H_3O^+$ reagent ion mode (Karl et al., 2012). In $H_3O^+$ reagent ion mode, both MVK and MACR undergo proton transfer and are detected as $C_4H_7O^+$ ($m/z$ = 71.050 Th), whereas in $NO^+$ reagent ion mode, ketones prefer the formation of clusters while aldehydes mostly undergo hydride ion transfer (Španěl et al., 1997). Therefore MVK is being detected as $C_4H_6O•NO^+$ ($m/z$ = 100.040 Th) and MACR as $C_4H_5O^+$ ($m/z$ = 69.034 Th) (Jud et al., 2016).

The instrument was connected to the chamber via a 3/8" stainless-steel sampling port with a flow rate of 10 slpm as described by (Schnitzhofer et al., 2014). Air was subsampled from this flow via a PEEK capillary (ID = 0.76 mm) at a flow



rate of 1.1 slpm from the centre of the tube in order to minimize wall contact of the sampled gas. The SRI-ToF-MS operated at a drift tube, capillary and inlet temperature of 40 °C to avoid thermal decomposition of ISOPOOH, a drift tube pressure of 2.3 mbar and a drift voltage of 550 V or 350 V. These conditions resulted in an E/N = 116 Td or E/N = 74 Td ($E$, electric field strength; $N$, number density of air in the drift tube; unit, Townsend, Td; 1 Td = $10^{-17}$ V cm$^2$) in $H_3O^+$ mode or $NO^+$

mode respectively.

The SRI-ToF-MS was characterised by regular calibration measurements. For this purpose, a standard gas mixture (Apel-Riemer Environmental, Inc., Broomfield (CO), USA; accuracy ± 5%) containing various pure and oxygenated hydrocarbons covering the whole mass range of the instrument (up to m/z = 205 Th) was dynamically diluted in zero air. Instrumental background was determined using zero air, which was generated by means of a Pt/Pd catalyst at 300°C from excess chamber

air, to allow instrumental background and calibration measurements at the same conditions as chamber measurements.

High resolution spectra were collected at a time resolution of 5 seconds and averaged over 5 minutes. The high resolution mass spectra data was analysed using the "PTR-ToF DataAnalyzer v4.40" software package (Müller et al., 2013).

### 2.3 LCU Measurements

A liquid calibration unit (LCU, IONICON Analytik) was used for calibration and characterisation of the behaviour of 1,2-

ISOPOOH, 4,3-ISOPOOH and $trans$-β-IEPOX standards with $H_3O^+$ and $NO^+$ as primary reagent ions inside the SRI-ToF-MS in laboratory studies at the University of Innsbruck. These laboratory studies are beyond the scope of this paper and will be described elsewhere in detail (Mentler et al., 2016).

### 2.4 Chemical Box Model

The MCM based chemical box model "University of Washington Chemical Model" (UWCMv2.2, (Wolfe et al., 2012;

Wolfe and Thornton, 2011)) was used for model simulations. The reactions were extracted from the MCM v3.3 (Saunders et al., 2003; Jenkin et al., 1997; Jenkin et al., 2015).

Model runs were initialised using the conditions of each experiment and constrained to isoprene, ozone, $NO_2$ and $SO_2$ concentrations, chamber temperature, relative humidity and actinic flux. The chamber pressure was assumed to be constant and pressure changes due to the adiabatic expansion during the campaign were excluded from the constraints. This is due to

the fact that said changes occur too fast (200 mbar over a few minutes) for the model to produce physically reasonable results.

## 3 Results and Discussion

### 3.1 ISOPOOH measurements using the SRI-ToF-MS

The deployed advanced SRI-ToF-MS allows a near artifact-free measurement of ISOPOOH with a specific fragmentation

pattern (Mentler et al., 2016). In contrast to standard instruments only about 13 % of the ISOPOOH isomers, 1,2-ISOPOOH



and 4,3-ISOPOOH, are detected as their respective conversion products MVK and MACR. About 87 % remain unconverted and are detected as ISOPOOH fragment ions. Both ISOPOOH isomers show a specific fragmentation pattern following ionisation, which can be used for identification. The signal at m/z = 85.066 Th ($C_5H_9O^+$) in $NO^+$ reagent ion mode, which is a prominent signal for both isomers, was used for identification and quantification of the total ISOPOOH concentration in

the experiment with a sensitivity $\varepsilon$ = 2 ncps ppbv$^{-1}$.

The goal of this CLOUD experiment was the investigation of SOA formation from dark ozonolysis of isoprene. Despite the lack of photolytically produced OH, oxidation of isoprene with OH plays an important role as a result of OH production from isoprene ozonolysis and due to the high reactivity of isoprene with the OH radical. Even though the experiments were conducted under low-NO conditions, only extremely low ISOPOOH mixing ratios could be observed in the experiments. To

understand why little ISOPOOH was observed and gain insight into the mechanistic reasons for this, measurements were compared to chemical box model simulations.

## 3.2 Model Simulations

### 3.2.1 Effect of OH Concentration on Discrepancy of Measurement and MCM predicted Product Concentrations

The chemical loss rates of ISOPOOH, ca.$5*10^6$ s$^{-1}$, only plays a minor role in comparison to residence time in the chamber

(see Table 2 for lifetimes of ISOPOOH) due to the low OH concentrations. The exchange rate of the chamber, $9.9*10^{-5}$ s$^{-1}$, is the main sink for all volatile organic compounds as well as low volatile organic compounds, if wall losses are ignored. Typical wall loss rates of condensable vapours such as sulfuric acid are about a factor 20 higher, on the order of $1.7*10^{-3}$ s$^{-1}$ (Almeida et al., 2013).

The chamber exchange rate is well known and depends only on the well-defined chamber inflow and outflow (2.8 h at

150 slpm). If the residence time is the dominant sink term, the concentration of reaction products only depend on their production rate and the chamber exchange rate. Product formation rates from ozonolysis of isoprene are also well known from the ozone and isoprene measurements. This leaves the concentration of the OH radical, which controls production of ISOPOOH, as the main uncertainty of the box model simulation due to a lack of direct OH measurements. Using sulfuric acid and $SO_2$ to derive OH concentrations, as done in previous CLOUD experiments, which relies on the sulfuric acid

production rate, was not possible in this study due to a direct source of sulfuric acid and $SO_2$ from injection of cloud condensation nuclei (CCN) consisting of liquid sulfuric acid. Therefore OH concentrations had to be estimated via model simulations which use the OH production rate from isoprene ozonolysis with a molar yield of 26 % (Malkin et al., 2010).

OH concentrations in the study were significantly lower than in the atmosphere since the experiments were carried out almost entirely in the dark, making isoprene ozonolysis the main source of the OH radicals.

When OH was not constrained in the model, [OH] = $0.5 – 1*10^5$ molecules cm$^{-3}$, the model was able to reproduce the general trend of the major oxidation products in the measurement. However, the model predicted ISOPOOH volume mixing ratios of several ppbv (Fig. 1, Fig. 2d), whereas m/z = 85.066 Th, corresponding to ISOPOOH with a limit of detection better





than 100 pptv, was only observed at very high isoprene concentrations ([Isoprene] > 150 ppbv) at a temperature of 273 K and the observed concentrations were very low. Comparison of the sum of the modelled ISOPOOH and observations thus shows a dramatically higher concentration (factor 20) of ISOPOOH in the model simulation. Comparison of MVK, MACR (Fig. 2b, 2c) and formaldehyde measurements with the MCM shows significantly lower concentration of these compounds in

the model simulations (Fig. 3, panel a-c) than observed. The difference makes up for 1.2-2 ppbv (50 %), 0.6 1.2 ppbv (25 %) and 6-10 ppbv (50 %) respectively.

To assess the impact of uncertainties in the model OH concentration, the production of the main oxidation products from reaction with OH and ozonolysis was studied as a function of OH concentration (Fig. 2) under experimental conditions. All compounds show a high dependence on OH concentrations. However, it was impossible to resolve the discrepancy between

model and measured first generation oxidation products by varying the concentration of OH radicals. While increasing OH by a factor of 3 leads to a good agreement of model and measurement for MACR, the simulation still shows lower concentrations for MVK (Fig. 2b, 2c). Further increase leads to an agreement in the case of MVK but to considerable higher concentrations of MACR compared to the measurement.

Additionally ISOPOOH concentrations increase significantly upon increase of OH concentration (Fig. 2d). This is in contrast

to observations, which showed extremely low ISOPOOH concentrations. It is highly  unlikely that the instrument would not detect an ISOPOOH signal at m/z = 85.066 Th at these concentrations that are significantly above the detection limit, nor any of the other characteristic ISOPOOH fragments, at the highest used OH model concentrations. These model runs highlight a problem in the mechanism, as high model OH concentrations are needed to improve agreement with MVK, MACR and formaldehyde, which however increases the model-measurement discrepancy for ISOPOOH.

**3.2.2 Inclusion of Wall Reactions in the MCM Model**

If ISOPOOH is significantly higher in the MCM based simulations but not destroyed inside the analytical instrument, as shown in laboratory studies, there has to be a sink for ISOPOOH in the experiment that is not included in the MCM but efficiently depletes the ISOPOOH reservoir. Decomposition of ISOPOOH to MVK, MACR and formaldehyde on metal surfaces inside metal tubing of analytical instruments is known from previous studies (Rivera-Rios et al., 2014).

$1,2 - ISOPOOH + stainless\ steel\ surface \rightarrow MVK + HCHO$                                                  (1)

$4,3 - ISOPOOH + stainless\ steel\ surface \rightarrow MACR + HCHO$                                                  (2)

Therefore the stainless steel surface of the CLOUD chamber cannot be neglected for these reactions. Thus reactions 1-2 that convert 1,2-ISOPOOH to MVK and formaldehyde and 4,3-ISOPOOH to MACR and formaldehyde were added to the MCM mechanism. Decomposition of other hydroperoxides is assumed to proceed in an analogous manner. All model simulations

that follow use OH concentrations as predicted by the model, which are one order of magnitude lower than typical average atmospheric concentrations (see Table 1) and range from $0.5 - 1*10^5$ molecules cm$^{-3}$.





Figure 3, panels d-f show the change of the modelled product concentration upon implementation of surface catalysed decomposition reactions of hydroxy hydroperoxides to smaller carbonyls on the walls of the chamber, which are known to take place upon contact with stainless steel. The decomposition reactions of ISOPOOH occurs isomer specific. The collision rate of ISOPOOH with the chamber walls is estimated to be $1.1*10^{-3}$ s$^{-1}$, in analogy to the previously determined loss rate of

condensable vapours in CLOUD (Almeida et al., 2013) and 100 % conversion efficiency per collision is assumed. Inclusion of these catalytic surface reactions resolves the discrepancy between model and measurement for MVK and MACR within the instrumental uncertainties. This is found for all concentrations and temperatures. The wall conversion is still effective at temperatures below 0°C. The discrepancy of a factor of 20 for isoprene hydroperoxides is not only caused by heterogeneous decomposition of ISOPOOH to MVK and MACR but also an additional unquantified formation of unsaturated C5-diols. The

latter are formed in surface reactions but were not included in this study since the different reagent ion modes ($H_3O^+$ and $NO^+$) did not allow for a conclusive identification due to interferences or signals below the limit of detection for these compounds.

The discrepancy between model and measurement in the case of formaldehyde is reduced. In the case of HCHO there is still a source equal to 5 ppbv (40 %) which is unaccounted for. Apart from instrumental uncertainties, this could be due to

uncharacterised wall production, not uncommon for formaldehyde in chamber experiments, or other surface catalytic reactions which have not been identified.

**3.3 Production, Loss and Chemical Pathway**

Inclusion of the non gas-phase heterogeneous wall reaction changes the relative importance of the reaction pathways for the main products. The production sources of MVK and MACR are shown in Fig. 4. The main production path for MVK shifts

from ozonolysis to an equal contribution of ozonolysis and heterogeneous production from 1,2-ISOPOOH. Ozonolysis remains the most important pathway for MACR but heterogeneous production from 4,3-ISOPOOH becomes the second most important and significant formation source. Even at low OH radical concentrations, the influence of the heterogeneous conversion of hydroperoxides on gas phase concentrations of carbonyls cannot be neglected.

Figure 5 shows the relative contribution of the specific sinks to the total loss of the respective ISOPOOH isomers during an

experiment set at 263 K. Without wall reactions, loss occurs mainly due to the gas exchange of the chamber which determines the residence time. Upon implementation of wall conversion reactions, the loss to the camber walls becomes the most important sink. Average lifetimes for each sink are given in Table . At low OH concentration ($< 1*10^{-5}$ molecules cm$^{-3}$) reaction with OH is negligible compared to residence time or wall loss. As long as the wall loss occurs fastest, 90 % of ISOPOOH will be irreversibly lost to the chamber walls and converted to MVK and MACR. Figure 6 shows the effect of the

conversion efficiency per wall collision on model measurement comparison. Even reducing the efficiency to less than 40 % gives a very good agreement of model and measurements within 5 % discrepancy while at the same time reducing the concentration of ISOPOOH significantly (factor 4). But collision efficiency is not the only factor in providing an opportunity for the catalytic decomposition to take place. The residence time on surfaces, most likely in the liquid layer due to





ISOPOOHs high hydrophilicity, is very long, and provides ample opportunity for the reaction to occur. The experiments were carried out at two different temperatures: -10°C and 10°C and different RH (60-95 %). During all experiments ISOPOOH, independent of temperature and RH, was irreversibly lost to the chamber surfaces and undergoes a decomposition reaction to form MVK, MACR and formaldehyde.

This study shows that heterogeneous reaction of ISOPOOH on the stainless steel chamber walls, likely via homolytic peroxy bond cleavage on contact with stainless steel surfaces as shown in Scheme 1, is consistent with the observed MVK, MACR and ISOPOOH concentrations. It has previously been shown that the reaction takes place on metal surfaces within GC-MS and standard PTR-ToF-MS instruments and on contact with heated metal surfaces (Nguyen et al., 2014b; Rivera-Rios et al., 2014) but also, as this study shows, on metal surfaces at temperatures below ambient. The heterogeneous reaction also

explains suppression of IEPOX formation, which in turn likely limits the formation of organic SOA and reactive uptake from isoprene oxidation in metal environmental simulation chambers.

## 4 Conclusion and Atmospheric Implications

This study shows that heterogeneous surface catalysed decomposition reactions of hydroxy hydroperoxides have a great impact on the gas phase composition in terms of the studied oxidation products such as carbonyls under low $NO_x$ conditions

when reactive peroxides are prevailingly formed. This is especially the case in stainless steel environmental simulation chambers.

In teflon chambers only a small if any conversion of ISOPOOH produced from low $NO_x$ isoprene oxidation can be observed (Rivera-Rios et al., 2014; St. Clair et al., 2016) . In stainless steel chambers on the other hand, as shown in this study, very effective decomposition of ISOPOOH leads to volatile C4 carbonyls and suppresses formation of the secondary products

IEPOX and isoprene dihydroxy dihydroperoxides that play an important role in SOA formation, uptake and growth. These processes cannot be explained by pure gas phase models. Implementation of additional heterogeneous surface reactions to the chemical mechanism is necessary to improve the agreement between model predictions and observations.

The decomposition may also be expected to occur for hydroxyl hydroperoxides that originate from different precursors. If a decomposition reaction takes place and the resulting products have different reactivity and physical properties, especially

volatility and hydrophilicity, particle formation and growth and properties may be misrepresented. Depending on the scientific question, catalytic surface reactions could lead to a misrepresentation of chemical processes in chamber studies in comparison to ambient measurements. This is especially important when single specific compounds are of fundamental significance rather than broader classes of compounds with comparable properties or if intermediates play an important role in the process of interest. In contrast, if processes of interest only occur via first generation products the conversion of

secondary products on walls is irrelevant. This distinction is often unknown and, therefore, it is essential to fully characterise and understand wall production and losses.




Surface reactions may not only play an important role in chamber experiments but also in the real atmosphere where plants, soil and aerosols may provide a large surface area for such reactions. These surfaces can potentially serve as electron donors and catalyse decomposition reactions in a similar manner as stainless steel environmental simulation chamber walls. Very recent experimental results show that the decomposition of ISOPOOH indeed occurs on plant surfaces (Canaval et al., 2016)

converting ISOPOOH to MVK and MACR and C5-diols. The fate of these compounds in relation to plant-surface atmosphere interactions is much more complex than in an environmental simulation chamber, where volatile decomposition products are released into the gas phase. It has already been shown that MACR is substantially taken up by tomato plants (Muramoto et al., 2015), while the fate of MVK yet remains uncertain. To understand and correctly interpret the ratio of ISOPOOH to MVK/MACR in the atmosphere, these processes have to be fully understood. Therefore it is important to

create experiments in the future that allow a better reproduction of real atmospheric conditions in terms of sinks and sources e.g. teflon chambers with real plants not only for the study of primary BVOC emissions but also to investigate the impact of plant surface reactions in altering gas phase composition of oxygenated volatile organic compounds

**Acknowledgements**

We would like to thank CERN for supporting CLOUD with important technical and financial resources, and for providing a

particle beam from the CERN Proton Synchrotron. This research has received funding from the EC Seventh Framework Programme (Marie Curie Initial Training Network "CLOUD-ITN" no. 215072, MC-ITN "CLOUD-TRAIN" no. 316662, and ERC-Advanced "ATMNUCLE" grant no. 227463), the German Federal Ministry of Education and Research (project nos. 01LK0902A and 01LK1222A), the Swiss National Science Foundation (project nos. 200020_135307 and 206620_130527), the Academy of Finland (Center of Excellence project no. 1118615), the Academy of Finland (135054,

133872, 251427, 139656, 139995, 137749, 141217, 141451), the Finnish Funding Agency for Technology and Innovation, the Nessling Foundation, the Austrian Science Fund (FWF; project no. P19546 and L593), the Portuguese Foundation for Science and Technology (project no. CERN/FP/116387/2010), the Swedish Research Council, Vetenskapsradet (grant 2011-5120), the Presidium of the Russian Academy of Sciences and Russian Foundation for Basic Research (grants 08-02-91006-CERN and 12-02-91522-CERN), and the US National Science Foundation (grants AGS1136479, AGS 1628530,

AGS1628491, AGS 1247421, AGS 1321987  and CHE1012293). We would like to thank Jean Rivera (Harvard University) for providing ISOPOOH standards and helpful discussion.

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



**Table 1: Average experimental conditions**

| # Run | Temperature [K] | RH[a] [%] | Isoprene[a] [ppbv] | Ozone[a] [ppbv] | OH [molecules cm$^{-3}$] | NO$_x$[b] [ppbv] | SO$_2$ [ppbv] |
|:---:|:---:|:---:|:---:|:---:|:---:|:---:|:---:|
| 1 | 283 | 93 | 40-250 | ~100 | ~$1*10^5$ | <0.25 | <0.5 |
| 2 | 263 | 80 | 60 | ~100 | ~$0.75*10^5$ | <0.25 | <0.5 |
| 3 | 283 | 90 | 100 | ~95 | ~$0.9*10^5$ | <0.25 | <0.5 |
| 4 | 263 | 70 | 100 | ~120 | ~$0.5*10^5$ | <0.25 | <0.5 |

[a]average values, [b]mainly NO$_2$



**Table 2: Lifetimes of the dominant ISOPOOH isomers**

| Compound | $\tau_{Dilution}$[a] | $\tau_{OH}$[b] | $\tau_{wall\ loss}$[c] |
|----------|---------------------|----------------|------------------------|
| 1,2-ISOPOOH | 2.8 h | 55.6 h | 15 min |
| 4,3-ISOPOOH | 2.8 h | 24.2 h | 15 min |

[a]at a flow of 150 slpm, [b]at [OH] = $1*10^5$ molecules cm$^{-3}$, [c]estimate





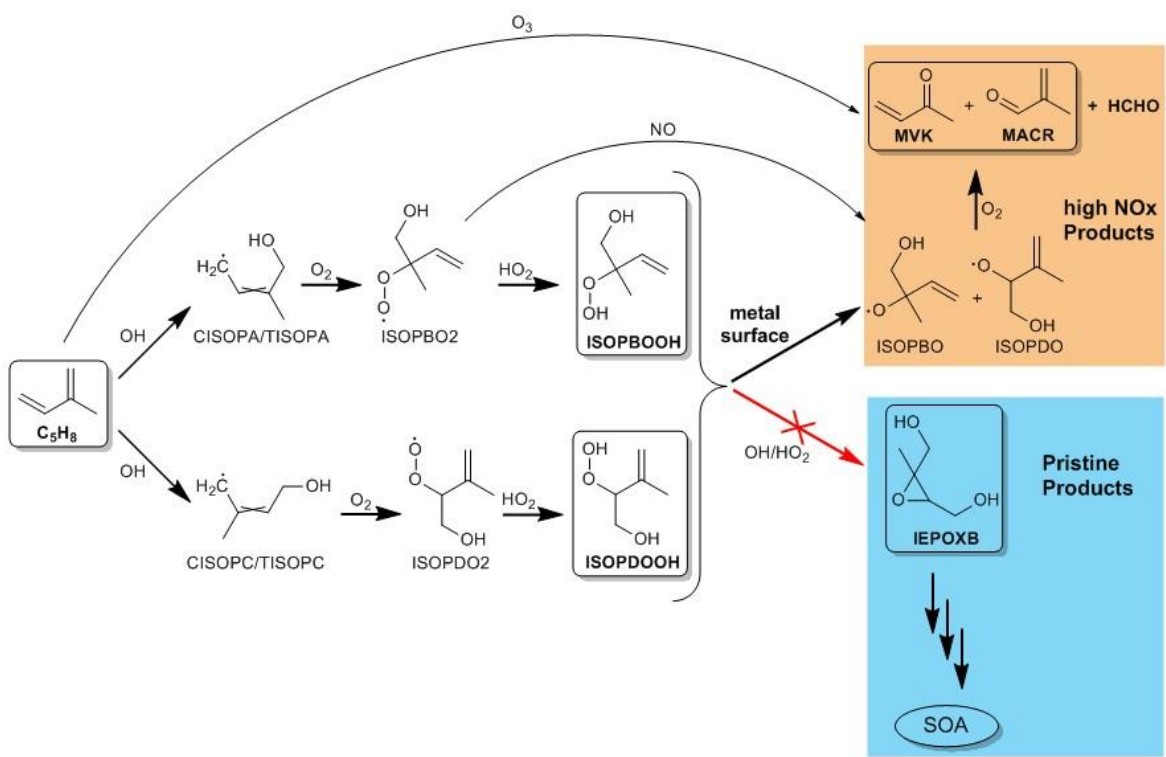

**Scheme 1: HO$_2$ dominated chemical pathway of the isoprene oxidation with OH based on the MCM v3.3. Isoprene oxidation in the presence of metal surfaces leads primarily to the same main oxidation products (MVK/MACR) as the high NO$_x$ pathway and catalytic surface reactions suppress the HO$_2$ pathway before the formation of β-IEPOX. MVK and MACR are also the main**
5 **oxidation products of isoprene ozonolysis.**





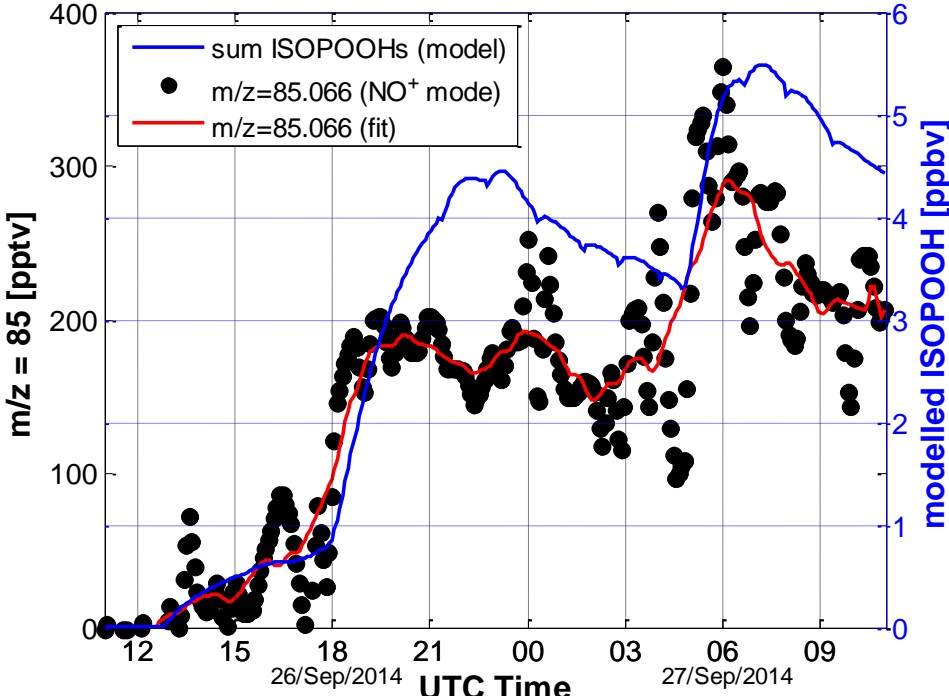

**Figure 1: Comparison of main ISOPOOH fragment in NO$^+$ reagent ion mode (C$_5$H$_9$O$^+$, m/z = 85.066) and MCM predictions at 1*10$^5$ molecules cm$^{-3}$ OH and up to 250 ppbv isoprene. The signal for the fragment follows the predicted trend but is about a factor of 20 lower than the expected ISOPOOH concentration in the model.**





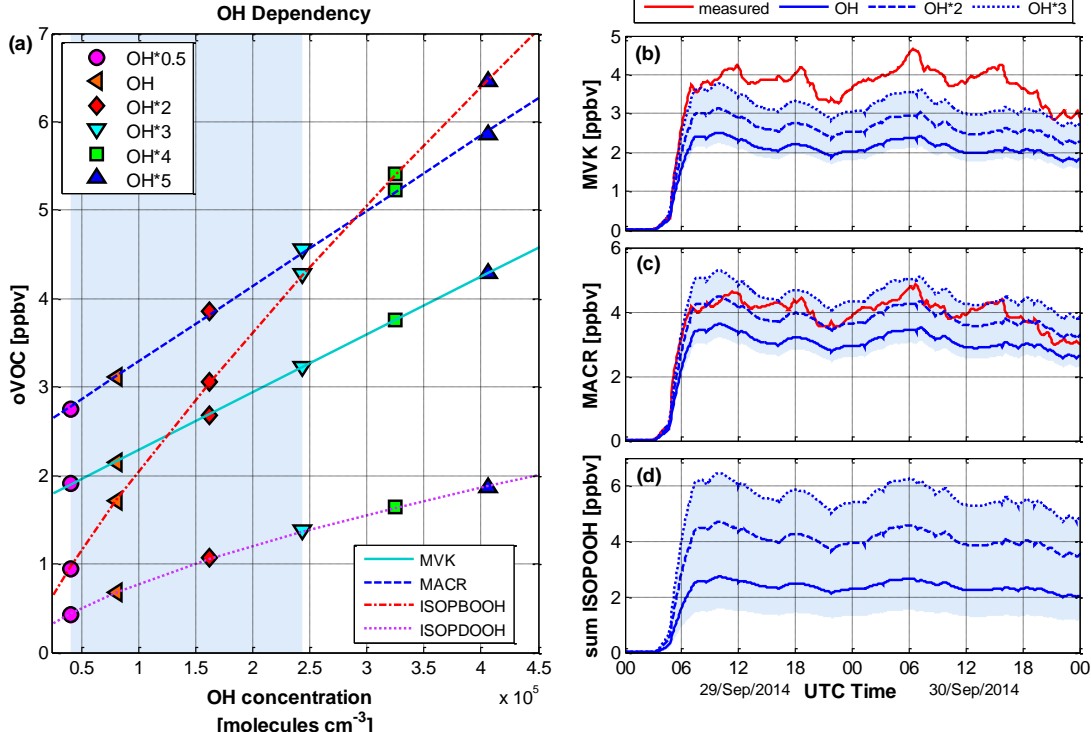

**Figure 2: a) OH dependence of the concentration of the main oxidation products at 263 K (Run2), b) changes in MVK prediction depending on OH concentration in comparison to measured data, c) changes in MVK prediction depending on OH concentration in comparison to measured data, d) changes in ISOPOOH prediction depending on OH concentration.**




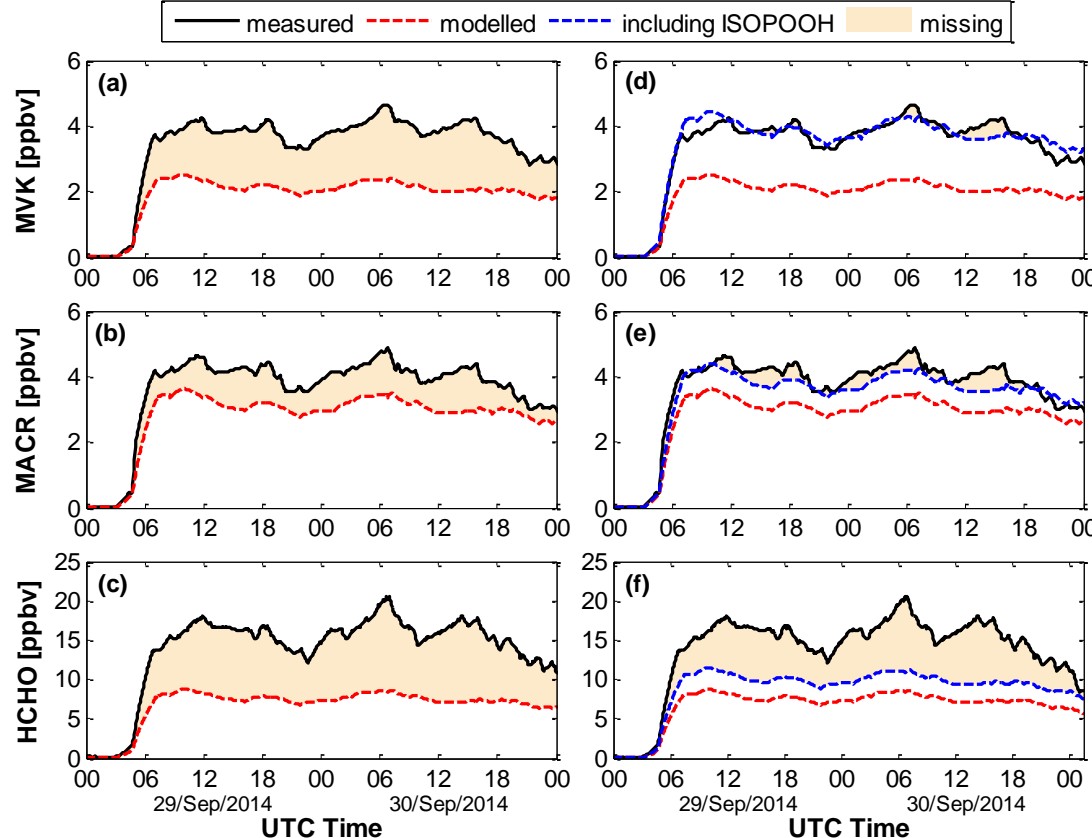

**Figure 3: Measured MVK, MACR and formaldehyde concentrations compared to model predictions. Panels a),b) and c) show MVK, MACR and HCHO in comparison to the model predictions. Modelled concentrations are by a factor of up to 2 lower. Panels d), e) and f) show the comparison to the model prediction after the inclusion of the respective ISOPOOH isomer decomposition. Inclusion of ISOPOOH conversion resolves the discrepancy for MVK and MACR. Dashed lines correspond to model simulations.**





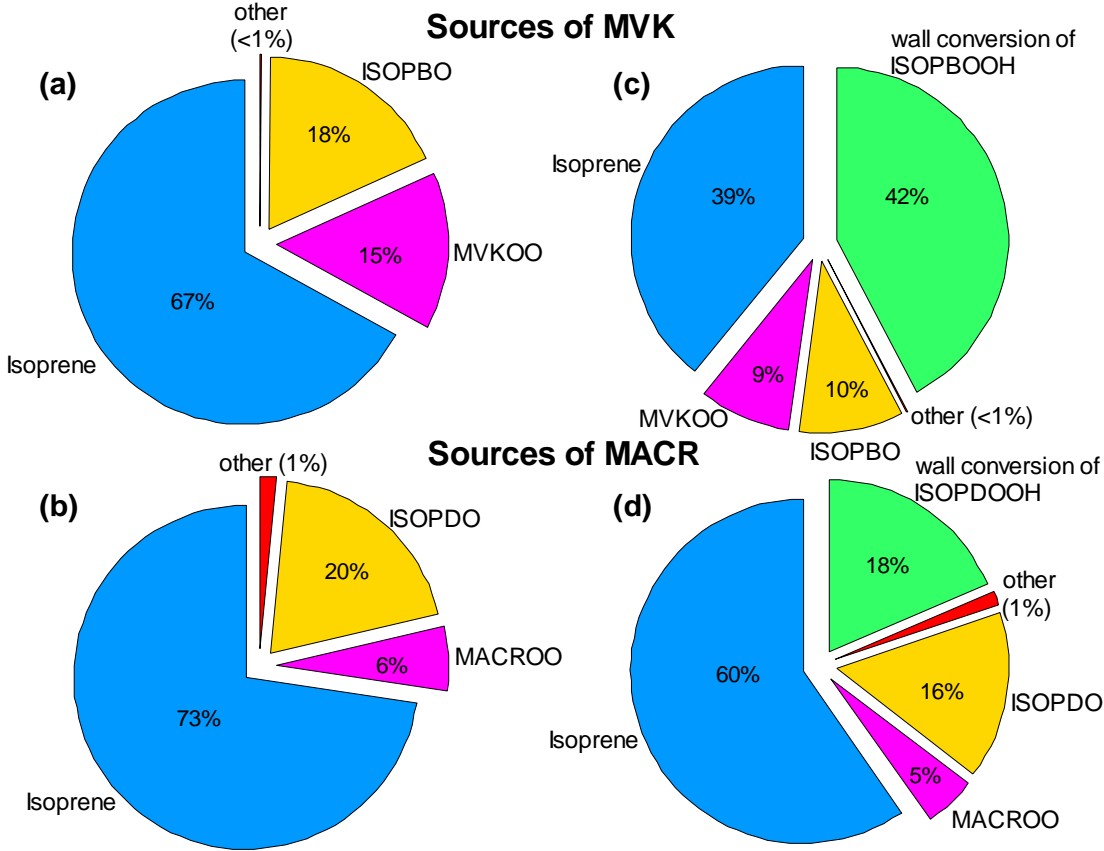

**Figure 4: Sources of MVK and MACR according to the relative importance of their precursors without (a,b) and with (c,d) additional wall conversion of ISOPOOH. Inclusion of the wall conversion of ISOPOOH to the model simulations changes the importance of the respective formation pathways. See Table S1 for explanation of source terms.**



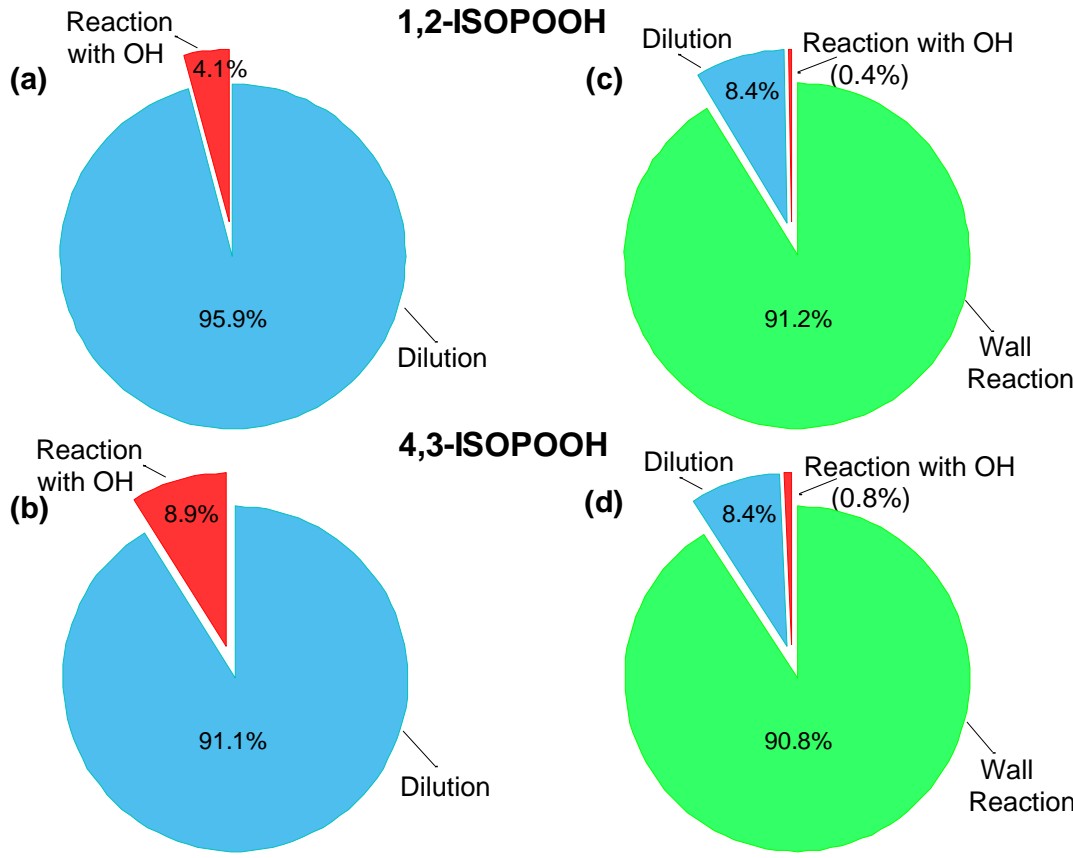

**Figure 5: Relative contribution of 1,2-ISOPOOH and 4,3-ISOPOOH sinks without (a,b) and with (c,d) additional wall reactions at an estimated wall loss rate of $1.1*10^{-3}$ s$^{-1}$. Loss due to reaction with OH is a negligible in comparison to wall loss and dilution at these low OH concentrations. More than 90 % of both isomers are being lost to the chamber walls.**





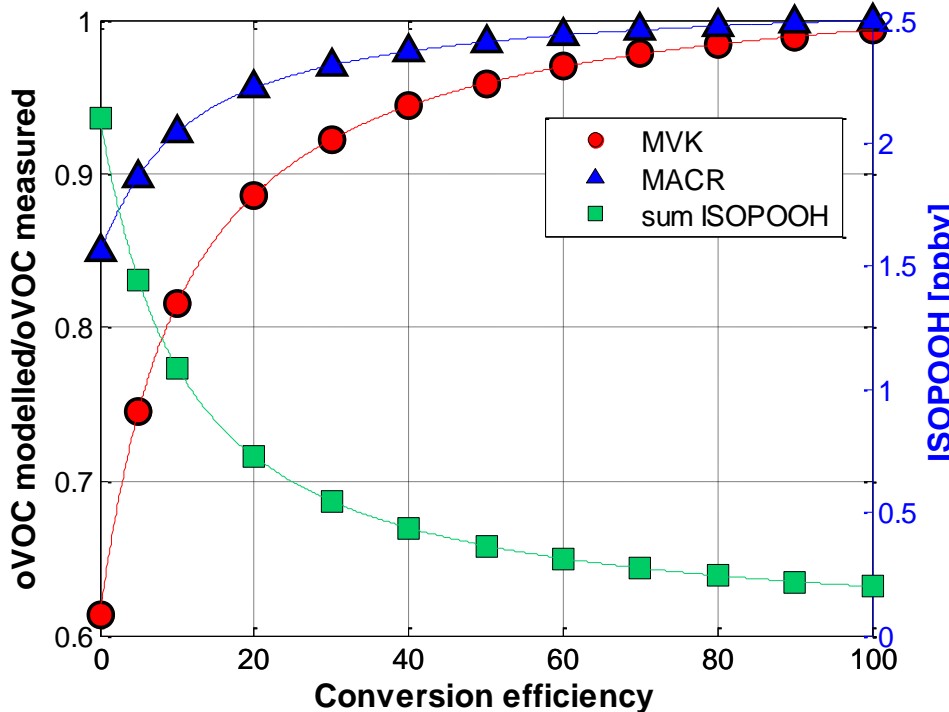

**Figure 6: Influence of conversion efficiency on the agreement of modelled VOC and measured VOC concentrations ranging from a conversion efficiency of 0 % (no wall reaction) to an efficiency of 100 % (complete decomposition on wall collision).**