# Peer review of "Technical Note: Conversion of Isoprene Hydroxy Hydroperoxides (ISOPOOH) on Metal Environmental Simulation Chamber Walls"

_Atmospheric Chemistry and Physics, 2016_

## Referee Comment (RC1) · Anonymous Referee #1 · 22 Nov 2016

This manuscript provides a straightforward explanation of the potential for biases in measurements of functionalized volatile organic compounds (VOCs) made in environmental chambers containing metal surfaces. It had previously been reported that instrumental setups, particularly those containing heated metal, could interfere with VOC measurements by providing sites for heterogeneous reactions to occur; here, the authors provide evidence that such reactions can occur on metallic chamber walls as well. In this study, the authors use selective reagent ion mass spectrometry to measure the VOC products of isoprene ozonolysis in a stainless steel chamber, and compare those measurements to simulated concentrations of the same VOCs using a box

model with chemistry from the Leeds MCM. Major discrepancies are found between measured and modeled concentrations of methyl vinyl ketone (MVK), methacrolein (MACR), formaldehyde, and ISOPOOH. Those discrepancies can largely be fixed by inclusion in the model of a decomposition reaction, in which ISOPOOH is catalytically converted on the chamber surface to formaldehyde and MVK or MACR (depending on the ISOPOOH isomer). The resulting conclusion - that decomposition on metal surfaces can represent a major loss process of ISOPOOH and can contribute markedly to increases in observed MVK and MACR concentrations in chamber experiments - is mostly insensitive to variations in assumptions in the model, such as OH concentrations and collision efficiency. It will be important to consider this effect and take it into account in future (and perhaps past) studies, and to investigate how it extends to other surfaces (e.g. plants), other experimental conditions (e.g. temperature and humidity), and other compounds (e.g. broader hydroperoxides).

General comments:

While the manuscript is qualitatively important in that it points out a previously unconsidered potential source of error in other studies involving hydroperoxides in metal chambers, in order for this to be quantitatively useful, it would be helpful to have some estimate of uncertainty. For example, correcting the product distributions of previous studies of isoprene photolysis or effectively designing new studies to avoid interferences in metal chambers will require estimates not just of the decomposition rate that best fit the measurements in the present study, but of the range of possible decomposition rates consistent with the measurements. Most useful, perhaps, would be upper and/or lower limits on the model parameters - e.g. collision efficiency and OH concentration - that might be extended to other studies. There are many elements where uncertainty enters into the results reported herein - modeled OH yields from ozonolysis, conversion efficiency of hydroperoxides on surfaces, measurement of compounds - and although some are treated individually in the manuscript (with Figures 2 and 6 showing how changing OH and conversion efficiency affect the model output), a more

comprehensive treatment of the aggregated uncertainty in the conclusions would make the results more broadly applicable, and aid in extending the findings to other studies.

Along those lines, connecting these results to other studies and other systems will be key. Are there previous studies - particularly photooxidation product studies - in which metal surfaces have been used before, such that previously reported results may be called in to question or require adjustment based on the findings detailed herein? If so, some mention of those cases in the conclusion and implication sections would be warranted. Additionally, are there related systems that involve hydroperoxides to which this work might apply?

Content comments:

Page 5, line 27: How does this OH production rate compare with other estimates, and what effect does that have on the model? Similarly, it might be useful to perform some sensitivity studies, changing your modeled yields of not just OH but MVK and MACR from isoprene + O3 (and isoprene + OH) and seeing how much that affects the results. It appears from a following paragraph (page 6, lines 7-13) and from Figure 2 that you performed some such studies; from those studies, can you conclude that the OH production rate used in the model was reasonable, or can you put any level of uncertainty on it?

Page 5, line 30: What does this [OH] number refer to? The modeled range of concentration present in the chamber when using the 26% yield number?

Page 7, line 5: The conversion efficiency per collision is another element in which it would be nice to look at the sensitivity of your results to the assumptions made. Figure 6 shows some of the model results if you loosen the assumption that conversion efficiency is not quantitative, but how do these values fit within the measurement uncertainty of MVK and MACR? How well does the modeled ISOPOOH match measurements? Using these values and the measurement uncertainty, can you estimate a range in which the conversion efficiency is likely to fall (or, perhaps more importantly, a

lower limit)?

Page 7, line 8-10: Do you have a reference for the formation of unsaturated C5-diols from ISOPOOH? In general, it would be nice to see the model-measurement discrepancy in ISOPOOH mixing ratios appear more in the results - say, in additional panels in Figure 3, or panel D of Figure 2. The formation of C5-diols may cause further discrepancy between measurements and models, but if so, that discrepancy should be quantifiable; if it points to an additional loss process, how does that compare to the processes shown in Figure 5?

Page 7, line 15: Do you have a reference for the claim that uncharacterized wall production of formaldehyde is common in chamber experiments? If it is common enough to have been quantified in the past, is it possible to estimate whether that source could account for the missing fraction you observe?

Page 7 line 32 - page 8 line 1: Do you expect there are conditions under which the residence time on surfaces would be too short for the reaction to occur efficiently, or under which the limiting factor in ISOPOOH decomposition would be something other than the collision rate?

Minor copyediting comments:

Page 1, line 27: "affection" should perhaps be "affecting"

Page 2, line 29: "ISOPOOH decomposes isomer specific to smaller carbonyls" is unclear.

Page 3, line 11: "26 m electro polished stainless-steel " should read "26 mˆ3 electropolished stainless steel"

Page 3, line 28: "being" is not needed

Page 3, line 31: parentheses not needed for this citation

Page 5, line 14: "rates" should be "rate" (or "plays" should be "play")

Page 6, line 5: What do you mean by "makes up for"? Was there a 1.2-2 ppbv (50%) discrepancy in the modeled and observed MVK that is now accounted for? Or do you mean the total difference between measured and modeled MVK is 1.2-2 ppbv? And is that 50% of the modeled or measured total? Also, missing a "-" between 0.6 and 1.2.

Page 7, line 3: "occurs" should perhaps be "are"

Page 7 , line 14: what exactly does the 40% refer to? Is 5 ppbv equal to 40% of the average total formaldehyde signal?

Page 8, line 1: "ISOPOOHs" should be "ISOPOOH's"

Page 8, line 3: tense disagreement between "was" and "undergoes"

---

## Author Comment (AC1) · 3 Feb 2017

We would like to thank the reviewer for taking the time to evaluate our manuscript and provide comments that have improved the manuscript. In the response below, the reviewer's questions are reproduced in blue and our replies are shown in black.

**Referee 1:**
General comments:
While the manuscript is qualitatively important in that it points out a previously unconsidered potential source of error in other studies involving hydroperoxides in metal chambers, in order for this to be quantitatively useful, it would be helpful to have some estimate of uncertainty. For example, correcting the product distributions of previous studies of isoprene photolysis or effectively designing new studies to avoid interferences in metal chambers will require estimates not just of the decomposition rate that best fit the measurements in the present study, but of the range of possible decomposition rates consistent with the measurements. Most useful, perhaps, would be upper and/or lower limits on the model parameters - e.g. collision efficiency and OH concentration - that might be extended to other studies. There are many elements where uncertainty enters into the results reported herein - modeled OH yields from ozonolysis, conversion efficiency of hydroperoxides on surfaces, measurement of compounds- and although some are treated individually in the manuscript (with Figures 2 and 6 showing how changing OH and conversion efficiency affect the model output), a more comprehensive treatment of the aggregated uncertainty in the conclusions would make the results more broadly applicable, and aid in extending the findings to other studies. Along those lines, connecting these results to other studies and other systems will be key. Are there previous studies - particularly photooxidation product studies - in which metal surfaces have been used before, such that previously reported results may be called in to question or require adjustment based on the findings detailed herein? If so, some mention of those cases in the conclusion and implication sections would be warranted. Additionally, are there related systems that involve hydroperoxides to which this work might apply?

It is hard to quantitatively translate this study to other studies as the surface effect can vary depending on, e.g., the experimental setup and the history of the previous experiments conducted in the same setup (number and manner of cleaning cycles between new sets of experiments, etc.). More information on uncertainties affecting this study has been provided in the comments below.

Even though the impact of the observed effect cannot be quantitatively transferred to other experiments, it is clear that metal surfaces are prone to this type of reactions. Metal surfaces not only affect peroxides but also affect observation of acids, and these are the main reason why photooxidation studies, especially under low NO$_x$ conditions, are most commonly carried out in PTFE chambers instead of metal chambers. However to fully understand the extent of decomposition rate and products of ISOPOOH on chamber walls or other compounds, it would be necessary to introduce the compound in question directly into the

chamber before the start of the experiment to fully characterise its behaviour in the chamber.

Content comments:
Page 5, line 27: How does this OH production rate compare with other estimates, and what effect does that have on the model?

Different approaches to determine OH concentration were tried to estimate the actual OH concentration:

1) Using $H_2SO_4$ and $SO_2$ concentrations as proxies for the OH concentration proved to be impossible due to artificial injection of $H_2SO_4$ and $SO_2$ from the injection of CCN from liquid sulfuric acid.

2) Calculating the OH concentration from the reacted Isoprene proved difficult due to the adiabatic expansions performed during this set of experiments which required constant changes of the chamber conditions (flows, temperature, pressure, RH), which were not necessarily well documented.

3) The OH yield given by the MCM v3.3 is a little higher (~33 %) compared to the literature OH yield of 26 %. The OH yield increases with increasing pressure (Malkin et al., 2010, Newland at al., 2015), the pressure during the CLOUD experiments was 5 – 200 mbar higher than the highest pressure observed in Malkin et al.

The yield provided by the MCM without additional adjustments provided the best agreement between model and measurement. Therefore it was used during this study.

Similarly, it might be useful to perform some sensitivity studies, changing your modeled yields of not just OH but MVK and MACR from isoprene + O3 (and isoprene + OH) and seeing how much that affects the results.

Modifying the yields of the reaction from isoprene + $O_3$ for MVK and MACR as well as the production of MVKOOA and MACROOA changes the product distribution as follows:

- Increasing the MACR yield from 30 % to 40 % and the MVK yield from 20 % to 50 % would be required to match model simulations to measurements. These modified yields are inconsistent with numerous laboratory experiments (Nguyen et al., 2016, Galloway et al., 2011 and references therein). While MACR would still be in the range of observed yields, even though at the upper limit, MVK would be more than 2 times higher than any observed yields. The impact on MVK is much more pronounced than the impact on MACR which is reasonable considering that 1,2-ISOPOOH is more abundantly formed than 4,3-ISOPOOH.

- Varying the yields for the $O_3$ + isoprene reaction does not resolve the issue of the formation of unobserved ISOPOOH. ISOPOOH concentration can only be matched to the measurements by either shutting down the isoprene + OH pathway or inclusion of wall decomposition reactions. Shutting down the isoprene + OH pathway is however unrealistic and inconsistent with known chemistry (26 %, Malkin et al., 2010, Newland at al., 2015).

It appears from a following paragraph (page 6, lines 7-13) and from Figure 2 that you performed some such studies; from those studies, can you conclude that the OH production rate used in the model was reasonable, or can you put any level of uncertainty on it?

There are three sources of uncertainty for the OH concentration in the experiment since OH could not be measured directly:

4) Photolysis of ozone. This is negligible as UV radiation was too weak and, in fact, turned off for most of the experiment.
5) Uncertainty of isoprene and ozone concentrations. Both were directly measured and isoprene was regularly calibrated at the same conditions as the chamber experiments (uncertainty < 5%).
6) Uncertainty of OH production in the MCM:

OH concentrations are more likely to be underestimated than overestimated, but the uncertainty is hard to quantify. Even an underestimation by a factor of 2 doesn't resolve the issue of the discrepancy between model and measurement of MVK and MACR nor does it resolve the issue of the unobserved ISOPOOH. On the contrary, while increasing OH reduces the discrepancy between model and measurements for MVK and MACR, it significantly increases the discrepancy between modelled ISOPOOH (increase with increasing OH) and the fact that barely any ISOPOOH was observed above the limit of detection. Therefore, the OH production rate as given by the MCM seems reasonable and in any case, it is not possible to reconcile the model under-prediction of MVK/MACR and model over-prediction of ISOPOOH by changing OH; figure2 demonstrates this, i.e., changes in OH affect both MVK/MACR and ISOPOOH in the same manner and thus they are not able to reconcile the measurement/model discrepancy.

Fundamentally, a change in reaction pathway, rather than a change in OH concentration is required, to explain the observations.

Page 5, line 30: What does this [OH] number refer to? The modeled range of concentration present in the chamber when using the 26% yield number?

[OH] is the concentration of OH in molecules $cm^{-3}$ from the MCM model simulation for experimental conditions and with the model only constrained to observed isoprene and ozone concentration.

Changed to:

The MCM model gives a range of $0.5 - 1*10^5$ molecules $cm^{-3}$ for the OH concentration, [OH], under the different experimental conditions and no further modelling constraints. These OH concentrations allow a reproduction of the general trend of the major oxidation products in the measurement.

Page 7, line 5: The conversion efficiency per collision is another element in which it would be nice to look at the sensitivity of your results to the assumptions made. Figure 6 shows some

of the model results if you loosen the assumption that conversion efficiency is not quantitative, but how do these values fit within the measurement uncertainty of MVK and MACR?

The uncertainty for the MVK measurements is on the order of 5%. The uncertainty for the MACR measurements is a little higher (~15%) due to the lack of a direct calibration.

Figure 2 has been updated to include the measurement uncertainties of MVK and MACR (red areas) as well as measured ISOPOOH:

[Figure]

**Figure 2: a) OH dependence of the concentration of the main oxidation products at 263 K (Run2), b) changes in MVK prediction depending on OH concentration in comparison to measured data, c) changes in MVK prediction depending on OH concentration in comparison to measured data, d) changes in ISOPOOH prediction depending on OH concentration. The shaded blue area corresponds to the range of OH concentrations and the red shaded area corresponds to the uncertainty in the respective measurements of MVK and MACR.**

The uncertainties in the quantification of MVK and MACR don't change the overall picture. While the uncertainty of the MVK measurement is of no effect, the uncertainty of the MACR measurement leads to a better agreement between model and measurement in the case of the inclusion of wall reactions. Without wall reactions, the discrepancy between model simulation and measurements cannot be explained by OH uncertainty or measurement uncertainty of the oxidation products.

Inclusion of the measurement uncertainties for MVK and MACR gives the following picture:

[Figure]

**Figure 6: Influence of conversion efficiency on the agreement of modelled VOC and measured VOC concentrations ranging from a conversion efficiency of 0 % (no wall reaction) to an efficiency of 100 % (complete decomposition on wall collision). The shaded red area corresponds to the uncertainty in the measurements of MVK.**

The uncertainty for MVK is of little effect regarding the overall results (see above). Even at around 40 % collision efficiency modelled and measured concentrations are in good agreement. The influence for MACR is more pronounced, as expected. If MACR concentrations were in fact lower on the lower range of the shaded area (higher MACR concentrations) an agreement higher than 85 % between model and measurement would be impossible. If they were on the higher side (lower MACR concentrations) an overestimation of the modelled concentrations would occur.

**How well does the modeled ISOPOOH match measurements?**

[Figure]

Comparing modelled ISOPOOH with measurements the discrepancy can be reduced from a factor of approx. 20 down to a factor of 2-3 upon inclusion of wall conversion reactions. The remaining discrepancy between model and measurements might be due to the measurement uncertainties in this regime close to the LOD and a lack of direct calibration during the CLOUD campaign (red area corresponds to a measurement uncertainty of 75%).

Using these values and the measurement uncertainty, can you estimate a range in which the conversion efficiency is likely to fall (or, perhaps more importantly, a lower limit)?

Looking at fig. 6 a conversion efficiency of as low as 20 % at a wall collision rate of $1.1*10^{-3}\,s^{-1}$ is sufficient to convert a significant amount (about two third) of ISOPOOH to MVK and MACR.

Page 7, line 30 changed to:
Figure 6 shows the effect of the conversion efficiency per wall collision on model measurement comparison. Even reducing the collision efficiency at a constant wall collision rate to less than 40 % gives a very good agreement of model and measurements within 10 % discrepancy and the measurement uncertainties while at the same time reducing the concentration of ISOPOOH significantly (factor 4). A conversion efficiency of as low as 20 % at a wall collision rate of $1.1*10^{-3}\,s^{-1}$ is sufficient to convert a significant amount (about two third) of ISOPOOH to MVK and MACR.

Page 7, line 8-10: Do you have a reference for the formation of unsaturated C5-diols from ISOPOOH?

The formation of C5-diols has been observed in laboratory studies in our group. The results will be published separately (Mentler et al., 2017). Unfortunately the C5-diol formation is not quantifiable using either $H_3O^+$ or $NO^+$ as primary reagent ions, which were used during the CLOUD 9 campaign, since they are not specific enough to rule out interferences.

In general, it would be nice to see the model-measurement discrepancy in ISOPOOH mixing ratios appear more in the results - say, in additional panels in Figure 3, or panel D of Figure 2. The formation of C5-diols may cause further discrepancy between measurements and models, but if so, that discrepancy should be quantifiable; if it points to an additional loss process, how does that compare to the processes shown in Figure 5?

Measured ISOPOOH is now included in an updated version of fig. 2 (see above). The measured concentrations were not included in the first version of the figure because the signal is below the LOD and no ISOPOOH could be observed during this experiment. ISOPOOH was only observed during the experimental run which is shown in Figure 1.

The generalized sink terms as presented in figure 5 should not be affected by the inclusion of C5-Diols. The importance of wall decomposition compared to any other process will remain

the same. What will change would be the product distribution from the decomposition process. So far the model assumes the isomer specific decomposition to MVK, MACR and formaldehyde only. We will refer to the manuscript by Mentler et al. for further reference.

Page 7, line 15: Do you have a reference for the claim that uncharacterized wall production of formaldehyde is common in chamber experiments? If it is common enough to have been quantified in the past, is it possible to estimate whether that source could account for the missing fraction you observe?

Several chamber cleanliness studies have been performed at the CLOUD chamber before the start of a new measurement campaign. Even in clean chambers a formation of small volatile organic compounds, especially formaldehyde can be observed. This emission of VOCs increases significantly upon addition of ozone to pure air (see Schnitzhofer et al, 2014). If this is the case for clean stainless steel surfaces, it is fair to assume that it will be an even more pronounced effect on stainless steel surfaces with an organic coating.
Similar to the question of conversion efficiency, this is hard to quantify as formation depends strongly on the surface area of the chamber and nature of the organic components on the chamber wall. The wall production is expected to change with the experimental setup and/or with a different composition of organic species. Furthermore the duration of the experiment needs to be taken into account. The history of previous experiments will also influence the composition of the organic layer on the chamber walls. Without knowing the exact nature of these organics, it is hard to determine the exact amount of wall production.

Page 7, line 15 changed to:
The discrepancy between model and measurement in the case of formaldehyde is reduced. In the case of HCHO there is still a source equal to 5 ppbv (40 % discrepancy compared to measurement) which is unaccounted for. Apart from instrumental uncertainties, this could be due to uncharacterised wall production, not uncommon for formaldehyde in chamber experiments, which is described for the CLOUD chamber in Schnitzhofer et al. (2014), or other surface catalytic reactions which have not been identified.

Page 7 line 32 - page 8 line 1: Do you expect there are conditions under which the residence time on surfaces would be too short for the reaction to occur efficiently, or under which the limiting factor in ISOPOOH decomposition would be something other than the collision rate?

Considering the differences in the chamber exchange rate ($9.9*10^{-5}\,s^{-1}$) and the wall collision rate ($1.1*10^{-3}\,s^{-1}$), even if the reaction wouldn't occur upon the first collision, there would be ample time for an efficient reaction to occur upon a later collision.
A saturation effect of the wall surfaces is unlikely to occur in the CLOUD experiments or generally experiments on this scale since only small concentrations are exposed to a high surface area.

Variation of the decomposition efficiency is shown in fig 6. Even comparatively low conversion efficiencies still lead to a strong decomposition of ISOPOOH. The same is true for the reaction time. Here, under these experimental conditions, a variation leads to no effect as long as it is faster than the chamber exchange rate.

The influence of wall decompositions decreases upon increase of OH concentration. But even after an increase of OH by approx. two orders of magnitude it remains the most important sink of ISOPOOH in this experimental setup.

Minor copyediting comments:
Page 1, line 27: "affection" should perhaps be "affecting" Done
Page 2, line 29: "ISOPOOH decomposes isomer specific to smaller carbonyls" is unclear.

Changed to:
The different ISOPOOH isomers decompose isomer specific to smaller carbonyls: 1,2-ISOPOOH forms MVK and formaldehyde and 4,3-ISOPOOH forms MACR and formaldehyde.

Page 3, line 11: "26 m electro polished stainless-steel " should read "26 mˆ3 electropolished stainless steel" Done
Page 3, line 28: "being" is not needed Done
Page 3, line 31: parentheses not needed for this citation Done
Page 5, line 14: "rates" should be "rate" (or "plays" should be "play") Done
Page 6, line 5: What do you mean by "makes up for"? Was there a 1.2-2 ppbv (50%) discrepancy in the modeled and observed MVK that is now accounted for? Or do you mean the total difference between measured and modeled MVK is 1.2-2 ppbv? And is that 50% of the modeled or measured total? Also, missing a "-" between 0.6 and 1.2.

Changed to:
The difference between model and measurements are 1.2-2 ppbv (50 %), 0.6-1.2 ppbv (25 %) and 6-10 ppbv (50 %) respectively.

Page 7, line 3: "occurs" should perhaps be "are" Done
Page 7 , line 14: what exactly does the 40% refer to? Is 5 ppbv equal to 40% of the average total formaldehyde signal?

Changed to:
The discrepancy between model and measurement in the case of formaldehyde is reduced. In the case of HCHO there is still a source equal to 5 ppbv (40 % discrepancy compared to measurement) which is unaccounted for.

Page 8, line 1: "ISOPOOHs" should be "ISOPOOH's" Done
Page 8, line 3: tense disagreement between "was" and "undergoes" Done

[revised manuscript text omitted]